The head turn paradigm to assess auditory laterality in cats: influence of ear position and repeated sound presentation

Konerding Wiebke S. 1
Zimmermann Elke 2
Bleich Eva 3
Hedrich Hans-Jürgen 3
Scheumann Marina marina.scheumann@tiho-hannover.de 2
1 Institute of AudioNeuro Technology and Department of Experimental Otology, Hannover Medical School , Hannover , Germany
2 Institute of Zoology, University of Veterinary Medicine Hannover , Hannover , Germany
3 Institute for Laboratory Animal Science and Central Animal Facility, Hannover Medical School , Hannover , Germany
Vallortigara Giorgio
Electronic publication date: 2017 Oct 24
Publication date: 2017
Volume: 5
Electronic Location ID: e3925
Received 2017 Aug 30; Accepted 2017 Sep 23
Copyright: ©2017 Konerding et al.
Copyright year: 2017
Copyright holder: Konerding et al.
License: This is an open access article distributed under the terms of the Creative Commons Attribution License, which permits unrestricted use, distribution, reproduction and adaptation in any medium and for any purpose provided that it is properly attributed. For attribution, the original author(s), title, publication source (PeerJ) and either DOI or URL of the article must be cited.
License URL: https://creativecommons.org/licenses/by/4.0/

Keywords: Orienting asymmetry, Head-turn pradigm, Auditory laterality, Acoustic communication, Arousal, Habituation, Cat, Mammal

Funding: University of Veterinary Medicine Hannover Hannover Medical School Konrad-Adenauer-Stiftung Deutsche Forschungsgemeinschaft This work was financially supported by the University of Veterinary Medicine Hannover, the Hannover Medical School and the Konrad-Adenauer-Stiftung: PhD-fellowship for WSK. Additionally, this publication was supported by Deutsche Forschungsgemeinschaft and University of Veterinary Medicine Hannover, Foundation within the funding programme Open Access Publishing. The funders had no role in study design, data collection and analysis, decision to publish, or preparation of the manuscript.

==============================
In most humans, speech is predominantly processed by the left hemisphere. This auditory laterality was formerly thought to be an exclusive human characteristic, but is now suggested to have pre-human origins. In studies on auditory laterality in nonhuman animals, the head turn paradigm has become very popular due to its non-invasive character. Although there are implications that the head turn direction indicates functional dominance of the contralateral hemisphere in processing a given sound, the validity of the paradigm is under debate. To validate the paradigm via comparison with imaging or electrophysiological methods, it is first necessary to establish turning biases at the individual level. Recently, the domestic cat, a common model in hearing research, has been found to show turning biases at the group level. To assess individual turning asymmetries in cats, we repeatedly presented kitten isolation calls and assessed whether differences in conveyed arousal changed the previously described left-wards lateralisation of conspecific vocalizations. Based on responses to 50 playback presentations (25 of high and 25 of low arousal), we calculated individual head turn indices. Based on the total data set, we found no consistent individual turning bias, irrespective of call category or sex of the receiver. Although the playback paradigm was chosen carefully to reduce any effects of lateralized loudness perception or changes in motivation due to habituation, individual head turn biases changed significantly in concordance with habituation to repeated playback-presentations and was predictable by small deflections in ear position prior to listening. When splitting the data set according to a decline in responsiveness after seven playback presentations, we revealed an initial left turning bias for most of our subjects (i.e., significant at the group level). We propose that this left turning bias is related to right hemisphere dominance in processes like vigilance behaviour or general arousal rather than on auditory processing, as such. Our findings suggest that both the experimental sequence and sound level differences, induced by asymmetric ear positions, strongly influence the outcome of the head turn paradigm and should be taken into account when evaluating auditory laterality at the behavioural level.

Introduction

A characteristic feature of human language is the auditory laterality (i.e., left hemispheric dominance) for speech processing which is supported by several behavioural and neurological studies (Dehaene-Lambertz, Dehaene & Hertz-Pannier, 2002; Fischer et al., 2009; Hugdahl & Westerhausen, 2015). To study preverbal laterality in human infants, Young and colleagues (1983; 1990) introduced the so-called head turn paradigm (or head-orienting-asymmetry paradigm). Thereby, a sound is played back from a loudspeaker 180° behind the subject. The unconditioned head turn response to the acoustic stimulus is suggested to indicate the dominance of the contralateral hemisphere in processing the auditory stimulus. Young and colleagues showed that human infants turned their head to the right when listening to speech sounds, suggesting that this paradigm is suited to reveal a left hemispheric dominance for speech processing.

Due to its non-invasive character and its simplicity, the head turn paradigm has become popular to study auditory laterality in non-human animals ranging from mammals (Gil-da Costa & Hauser, 2006; Hauser & Andersson, 1994; Leliveld, Scheumann & Zimmermann, 2010; Scheumann & Zimmermann, 2008), to birds (Palleroni & Hauser, 2003), to amphibians (Xue et al., 2015). For example, in rhesus macaques (Hauser & Andersson, 1994) and sea lions (Böye, Güntürkün & Vauclair, 2005) a right ear preference for conspecific versus heterospecific calls was shown, suggesting a left hemispheric dominance for processing conspecific vocalisations similar to human speech processing (for review see Ocklenburg, Ströckens & Güntürkün, 2013; Taglialatela, 2007). However, the outcomes using the head turn paradigm are not consistent and the interpretation of the paradigm is further complicated as the head turn asymmetries in response to species-specific calls are influenced by a variety of different intrinsic and extrinsic factors. For example, specific acoustic properties of the calls (Ghazanfar, Smith-Rohrberg & Hauser, 2001; Hauser, Agnetta & Perez, 1998; Ratcliffe & Reby, 2014; Siniscalchi et al., 2012), familiarity with the sender (Basile et al., 2009; Basile, Lemasson & Blois-Heulin, 2009; Lemasson et al., 2010), characteristics of the sender (Leliveld, Scheumann & Zimmermann, 2010; Scheumann & Zimmermann, 2008) or emotional valence (Basile, Lemasson & Blois-Heulin, 2009; Reinholz-Trojan et al., 2012; Xue et al., 2015) alter the species-specific asymmetries in response to conspecific calls. Thus, it is argued that the asymmetries revealed by the head turn paradigm might be related to the induced arousal, based on the conveyed behavioural relevance of a given sound. This is partly consistent with findings in humans, as speech laterality can be shifted away from the typical left hemispheric dominance when listeners focus their attention on the emotional (i.e., prosodic) information of speech rather than its linguistic content (Ley & Bryden, 1982). The head turn paradigm might therefore be more suited to revealing lateralization due to induced arousal rather than being an indicator for lateralization of species specific calls, comparable to human speech lateralization.

The validity of the paradigm has been challenged by a study in adult humans, which did not show the expected right ear turning bias in a supermarket environment (Fischer et al., 2009). The authors concluded that factors like attention and experimental situation significantly influence the outcome of the head turn paradigm (Fischer et al., 2009). Thus, further methodological assessments will be necessary to prove the validity of the paradigm. To validate the paradigm, it would be necessary to compare individual turning-biases to brain imaging or electrophysiological data derived from the same individuals (Fischer et al., 2009). However, to the best of our knowledge, no study has presented statistical evidence for individual head turn biases. The reason for this is that authors used only a low number of stimuli due to their concern that animals would stop responding after only view repetitions (Hauser & Andersson, 1994; Reinholz-Trojan et al., 2012; Teufel, Hammerschmidt & Fischer, 2007) which makes individual analysis impossible (Leliveld, Scheumann & Zimmermann, 2010; Lemasson et al., 2010; Siniscalchi, Quaranta & Rogers, 2008). Lemasson et al. (2010) provided a study where each sound was presented 40 times, which indicates that individual head turn biases may be achieved. However, the authors did not report individual asymmetries and also did not comment on potential habituation effects. Thus, the aim of our study was to derive individual head turn biases towards species-specific calls of different arousal and to investigate the effect of repeated sound presentations for the domestic cat. The cat is a common model in hearing research (e.g., Eggermont, 1998; Heffner & Heffner, 1985; Land et al., 2016) and shows lateralized motor behaviour (Konerding et al., 2012; Wells & Millsopp, 2009; Wells & Millsopp, 2012).

Recently, Siniscalchi and colleagues (2016) published that cats turn their head to the right in response to conspecific calls, indicating the typical left hemispheric dominance in processing species-specific calls. However, only purring and meow sounds elicit right ear preferences, whereas growls showed a tendency in the opposite direction. The authors argued that this might be explained by different emotional meaning (Siniscalchi, Laddago & Quaranta, 2016). Growls might cause a shift to the right hemisphere due to higher levels of induced arousal or behavioural relevance. Nevertheless, the acoustic properties of these call types are very different (noisy to tonal calls), which might also affect hemispheric processing. Thus, to address the impact of induced arousal on auditory laterality in response to species-specific calls, it would be useful to investigate auditory laterality in response to the same call type recorded in low versus high arousal states.

In our previous publications, we already showed that kitten isolation calls show changes in acoustic properties during low versus high arousal behavioural conditions (Scheumann et al., 2012) and that these differences in conveyed arousal induced concordant differences in responsiveness in female cats (Konerding et al., 2016). Thereby, females oriented significantly faster towards the sound-source to high versus low arousal kitten calls. To assess the impact of induced arousal on the lateralization of conspecific calls, we investigated potential differences in head turn direction toward these two call categories. These calls did only differ in a defined set of acoustic parameters, related to the arousing property of the eliciting behavioural situation (Scheumann et al., 2012) and have been found to influence the urgency to respond in female but not male cats (Konerding et al., 2016). We expected a shift to a left ear preference (i.e., more left than right head turns) for high compared to low arousal calls in female cats only. As males showed no behavioral differences between either arousal categories (Konerding et al., 2016) we expect them to show the typical right ear/left hemispheric dominance to both types of this species-specific vocalization.

As we were the first to assess individual head turn responses, we decided to also look at potential influencing factors in more detail. The first factor is the effect of overall habituation to the playback-situation. This can be expected to have a greater impact on the results in an individual based approach than when combining several animals, which are tested only once per call type. The second is related to our animal model. As domestic cats have highly movable pinnae that are sometimes switched back and forth within milliseconds, we were concerned with potential loudness differences between the ears, based on ear position. As we took care to reduce the risk of habituation and monitored the ear position to choose the best time-point for playback-presentation, we expected these factors to have only minor impact on the result of our experiments.

Material and Methods

Subjects

We tested 15 adult cats (eight males, seven females) aged one to eight years (meanm = 2.6, meanf = 3.3). Housing conditions were the same, as described in Konerding et al. (2016). All subjects were not neutered/ castrated and originated from and were kept at the breeding facility of the Central Animal Facility of the Hannover Medical School. Adult cats lived in same-sex groups of 2–5 individuals, with changing composition based on breeding schedules. The cats were kept indoors in a controlled environment (light-dark cycle 12:12, 22 ± 2 °C, 55 ± 10% humidity). The rooms (12.5 m2–20.6 m2, height: 2.6 m) were enriched with wooden boxes, tables and shelves, plastic toys and bars for scratching. As an additional heat source, each room was equipped with an infrared lamp. The cats were fed daily with tinned (Whiskas® tins; Mars GmbH, Verden, Germany) and dry cat food (SDS Pet Food; Special Diets Services, Witham, Essex, UK) and were provided with water ad libitum. The animal husbandry fulfilled all recommendations for domestic cats as required by the guidelines of the European Union (ETS 123, Directive 2010/63/EU) and was approved by the local veterinary authority (No. 42500/1H).

Preparation of playback stimuli

The playback stimuli originated from a previous study on the acoustic communication of the domestic cat (Konerding et al., 2016). They were recorded from kittens aged nine to 11 days during separation conditions, varying only with regard to the amount of handling, either without (low arousal) or with (high arousal) handling by a human observer (for recording details see Scheumann et al., 2012). We used 14 calls from 7 senders as playback stimuli: one call of low and one of high arousal from each sender (Fig. 1). Multi-parametric sound analyses revealed that specific acoustic cues of the kitten isolation call (i.e., call duration, voicing and fundamental frequency) are distinct with regard to the arousal state of the sender (paired t-tests: p ≤ 0.033; Fisher Omnibus test: χ2 = 44.7, df = 20, p = 0.001; Scheumann et al., 2012). The stimuli were played back at a standardised sound pressure level (SPL) of 70 ± 2 dB at a distance of 1.8 m (i.e., licking distance, see below) from the loudspeaker (quadral Argentum 02.1, quadral GmbH & Co. KG; RMS fast measurement: Brüel and Kjær 2610, highpass filter: 22.4 Hz), to match the loudness of natural kitten vocalisations (Romand & Ehret, 1984).

Figure 1 Representative low (A) and high (B) arousal kitten calls of the same sender.

Depicted are oscillograms and spectrograms of each call from Konerding et al. (2016).

Playback experiments and experimental set-up

A subset of the behavioural recordings has already been analysed to assess the impact of conveyed arousal on the latency to orient towards the loudspeaker (Konerding et al., 2016). Thus, the methods have already been described in detail. In short, the cats were tested in a sound attenuated surrounding. The experimental cage was equipped with a drinking bottle containing a milk/water mixture and at the opposite side the loudspeaker was placed behind an opening in the foam of one movable wall (Fig. 2). Thereby, sounds were perceived from the defined direction (180° behind) and distance (1.8 m) while the cat was within licking distance. Before the actual experiments started, each animal got accustomed to the experimental set-up.

To reduce the risk of habituation, we limited the number of stimuli presented per trial and the number of trials per week. Each cat was tested individually 2 to 4 times a week and usually (61%) only one stimulus (maximum 4) was played during each trial. When more stimuli were played, the inter-stimulus interval was at least 1 min. The stimuli were played back in pseudo-randomised order. The stimulus was initiated by the experimenter based on videographic monitoring of the animal, i.e., when the subject was within licking distance of the drinking bottle, with its body aligned closely to the bottle-loudspeaker axis and its head held approximately straight (cage bars served as visual reference). The freedom for head movements was restricted by additional wire mesh to an angle of ±30 degree (Fig. 2). As the cat has highly movable pinnae (Populin & Yin, 1998), we took care that not only the head was straight but also the ears had the same distance to the loudspeaker when presenting a given call, i.e., the tips of the pinnae were parallel to the bars of the test cage. Still, we could not avoid small deflections (see video analysis) in the ear position during listening and included this as a potential confounding factor in the statistical analysis. The experiments of one subject were completed when 25 stimulus presentations of each category (i.e., 25 low and 25 high arousal calls) had been scored in the video analysis.

Figure 2 Schematic drawing of the playback set-up.

Depicted is a cat inside the experimental cage, in front of a drinking bottle. The cage is equipped with two wire meshes, restricting the freedom of head movements. The cage is surrounded by sound attenuating foam and the loudspeaker is placed inside an opening 180° behind the subject. The playback onset is indicated by the flashing of a diode (red). Cage bars were used as reference for parallel position of the tips of the pinnae. In this example pinnae showed as a small deviation from parallel to the left.

Video analysis

The video analyses were performed blind to the respective playback stimulus (i.e., without acoustic information). Thus, all experiments were scanned for playback presentations indicated by the lightening of a diode (invisible to the subject). We analysed only those presentations where a subject was in drinking distance of the bottle at the first flashing of the diode. We scored the first head turn response within 5 s of stimulus onset (Interact 32 software, Version 8, Mangold, Arnstorf, Germany; e.g., Basile et al., 2009; Siniscalchi et al., 2012; Siniscalchi, Quaranta & Rogers, 2008). The ear position prior to listening to the calls was assessed based on any deflections from a parallel position, using the cage bars as reference (see Fig. 2). We noted whether the left or the right pinna was directed more closely to the loudspeaker at the first flashing of the diode (i.e., defined as prior to listening).

To test for inter-observer reliability, 25% of the playback presentations (i.e., complete data of two males and two females) were reanalysed by a second observer. The comparison of the indices (see Statistical analysis) for both the head turn direction and the ear position derived by the two observers correlated significantly, indicating a high inter-observer reliability (two-tailed Pearson correlation, linear regression: head turn: p = 0.02, r = 0.98, y = 0.02 + 0.83x; ear position: p < 0.01, r = 0.10, y =  − 0.01 + 1.00x).

Statistical analysis

We calculated for each individual separately (for raw data see supplement Table S1), the head turn index (HTI) from the numbers of left (L) and right (R) head turns as HTI = (R − L)/(R + L). The HTI ranges from −1, for exclusive left bias, to +1, for exclusive right bias, with 0 indicating equal distribution of left and right turns. Individual turning biases were assessed via an exact binomial test. Accordingly, we defined individuals with a significant bias as left (negative HTI) lateralised, right (positive HTI) lateralised, or non-lateralised (p > 0.05). HTIs were assessed for the two call types and the two sexes separately. Asymmetries at group-level were assessed via two-tailed, one-sample t-tests, to report significant deviations from zero. To investigate potential confounding factors, we performed a binomial GLMM (Zuur et al., 2009) to assess the influence of the explanatory variables: Sex (male/female), Arousal (high/low), Order (playback presentation #1–50) and Initial Ear Position (straight, right, left) on the test variable Head Turn Direction (right, left) and used Subject as random variable.

Non-linear regressions using a one-phase decay function were calculated describing the effect of repeated sound presentation on the number of responding individuals and the HTI (GraphPad Prism 5). By splitting the data set when only about 50% of the subjects were responding to the playback presentation (see Results), we assessed individual HTIs for both the first seven and the following seven presentations. Based on these HTIs, we assessed whether potential group-level biases in head turn responses changed based on habituation (Wilcoxon signed rank test). Biases at individual-level could not be statistically assessed after splitting the data set, due to the remaining small sample size. For comparison with other head turn studies, we also assessed group-level HTIs, based on the first head turn response of each individual to each of the two call categories.

To investigate the potential influence of ear position prior to listening to the playback calls we analysed whether the prevalence of same-side turns was significantly higher than expected by chance (t-test vs 50%) and separately computed HTIs only for those occasions where the ear position was straight.

Prior to each analysis, we checked normality using the Kolmogorov–Smirnov test (normality was assumed if p > 0.05) and used non-parametric versions of the test statistic, whenever applicable. The level of significance was set at p < 0.05 and whenever available, we performed exact versions of the test statistics.

Results

Head turn bias

On individual-level, two cats (one male, one female) showed a significant head turn bias in response to low arousal calls, only (female: HTI = 1.00, n = 7 head turns; male: HTI = 0.78, n = 9 head turns, Fig. S1). Concordantly, there was no significant group-level bias based on individual HTIs, irrespective of sex and arousal (Table 1). Also, when considering all 50 stimulus presentations together, we found no significant group-level bias (mean = 0.091; t-test: t = 1.228, p = 0.240).

Table 1 Individual head turn indices did not differ significantly from chance at the group level.

Sex	Arousal	Mean HTI	t-value	p-value	
Male	Low	0.186	1.432	0.195	
High	0.139	2.252	0.059	
Female	Low	0.114	0.643	0.544	
High	−0.049	−0.368	0.726	

On investigating possible methodological confounding factors, a binomial GLMM analysis revealed that the direction of head turns was significantly influenced by Order (z-score = 2.14, p = 0.032) and Initial Ear Position (position right: z-score = 4.97, p < 0.001, position straight: z-score = 1.80, p = 0.072), but not by Sex (z-score = −0.585, p = 0.559) and Arousal (z-score = 0.692, p = 0.489). On calculating a further binomial GLMM, adding the interaction term Order*Initial Ear Position, we found no significant interactions between these two factors (z-score < 0.551, p ≥ 0.582). Thus, we further analysed the two influencing factors separately.

Habituation effect

Non-linear regression analysis of the head turn response using the one-phase decay function revealed a decline below 50% after the first seven stimulus presentations which levelled off at around 32.45% (i.e., 5 responding subjects) without converging to zero (non-linear regression: r2 = 0.453, Y0 = 89%, plateau = 32.45%, K = 0.167). In contrast, the distribution of the HTI across stimulus presentations showed an inverse progression (Fig. 3). Thus, the one-phase decay function increased for the first 7 stimulus presentations and reached its plateau at a HTI of 0.15 (r2 = 0.09, Y0 = −0.77, plateau = 0.15 K = 0.243).

Figure 3 Influence of habituation on the number of responding subjects and the head turn index (HTI).

Given are the change of responding subjects (blue) and the HTI (red) during 50 playback presentations. The regression line (one-phase decay) indicates a habituation (50% responding subjects) within the first 7 presentations (grey box) which is accompanied by a zero-crossing of the HTI-regression line.

To further characterise the effect of habituation, we separately calculated the individual HTIs for the first seven presentations and the following seven playback presentations (Fig. 4). The low number of responses did not allow for a statistical analysis on an individual-level. The analysis for all 15 subjects revealed significant group-level biases to the left for the first seven presentations (negative HTIs; Wilcoxon test vs 0: W = 16, p = 0.038) but no significant group-level bias for the following seven presentations (Wilcoxon test vs 0: W = 50.5, p = 0.097). Thereby the HTIs significantly shifted to the right when comparing the first with the following seven presentations (Wilcoxon test: W = 10.5, p = 0.008).

Figure 4 The head turn indices (HTI) based on the first seven, but not on the following seven, playback presentations shows a group-level bias to the left.

Depicted are individual data points (circles) connected by lines and the respective group medians (stars). Medians were chosen, as the data of the “following 7” were not normally distributed. **Wilcoxon signed rank test: p = 0.008; *Wilcoxon vs. 0: p = 0.038.

Concordantly, when only considering the very first head turn response of each individual and calculating HTIs at the group-level, we found more left than right head turn responses, with negative HTIs for all, but one, subgroups. The biases were however moderate (median absolute value: HTI = −0.27) and did not differ statistically from chance (Table 2).

Ear position

We used a very conservative method to define ear position asymmetries; thereby every visible deflection from a parallel ear position was noted (Fig. 2 indicating a small deviation from parallel to the left). In total, there were only 45 cases in which a head turn response was preceded by a parallel ear position, in 82 cases the right ear was closer to the loudspeaker and in 163 cases the left ear was closer to the loudspeaker. In an average 64% of cases a head turn was preceded by a same-side ear position; this was significantly higher than expected by chance (t-test vs 50%: t = 3.741, p = 0.002; Fig. 5). When calculating individual HTIs based only on the occasions when the ears were parallel, the HTIs did not differ significantly from zero (Wilcoxon test vs 0: W = 60.50, p = 0.281).

Table 2 First head turn response to low and high arousal kitten calls.

		Left turn	Right turn	HTI	p-value	
Male	Low	3	5	0.25	0.727	
High	7	1	−0.75	0.070	
Total	10	6	−0.25	0.454	
Female	Low	5	2	−0.43	0.453	
High	4	3	−0.14	1.000	
Total	9	5	−0.29	0.424	
Both	Low	8	7	−0.07	1.000	
High	11	4	−0.47	0.118	
Total	19	11	−0.27	0.200	

Figure 5 Influence of initial ear position on the direction of the head turn.

Depicted are means (N = 15) and standard deviations (whisker), with individual data points (circles). ∗∗one-sample- t-test vs. 50%: p = 0.002.

Discussion

The aim of our study was to reveal auditory laterality in domestic cats at individual-level in responses to repeated playback-presentations of kitten calls conveying either low or high arousal. Only two cats (one male, one female) showed a significant turning bias in response to low arousal kitten calls. Thus, we revealed neither consistent individual turning biases nor laterality at group-level, irrespective of sex or arousal category. Only, when splitting the data set based on habituation, we could reveal a head turn asymmetry at group-level. Overall, subjects turned more to the left side during the first seven presentations, when responsiveness was still high. Afterwards, the subjects turned more to the right side. To the best of our knowledge, we are the first to report a shift in head turn asymmetries during habituation to repeated sound presentations. However, findings have to be considered with caution as we also revealed a significant influence of the ear position prior to responding, with about 64% of the head turn responses being preceded by a same-side ear position (chance level: 50%). The result of a binomial GLMM showed that both factors did not interact with each other, thus, in the following, we will discuss these two independent factors separately.

Habituation effect

The number of responding subjects significantly declined in the course of the experiments and only within the first seven presentations did more than 70% of subjects respond with a head turn, the number subsequently declining to about 35%, without converging to zero. When dividing our dataset after the regression line dropped below 50% responding subjects, we found opposite turning biases at group-level. Whereas initially the head turn was to the left, which based on the rationale of the head turn paradigm would indicate a higher involvement of the right hemisphere, the HTIs based on the following seven presentations were shifted to the right (not reaching a significant group-level bias). The initial right turning bias was confirmed when only considering the very first response of each subject, the results were however not significant. The lack of statistical power is based on the relatively small sample size, as the study-design was chosen to assess individual biases, rather than to reveal asymmetries at the group-level based on only on response per individual.

Although a habituation effect on the head turn asymmetry has not been reported so far, conflicting results in domestic dogs might be interpreted in this way. Presenting only one call per subject, Reinholz-Trojan and colleagues (2012) revealed left turning biases (indicating right hemispheric dominance) towards dog barks. On the contrary, Siniscalchi and colleagues (2012; 2008), who assessed HTIs based on repeated (n = 7) playback presentations, revealed right turning biases towards dog barks. The authors excluded the subsequent playback presentations, due to habituation to the playback stimuli, using a similar approach as in our study. The discrepancy between the results of the two research groups may indicate a decreased dominance of the right hemisphere and an increased involvement of the left hemisphere during habituation to repeated presentations of species-specific calls in dogs. If so, the process would already influence the head turn direction prior to an observable decline in responsiveness (i.e., percentage of stimuli that induced a head turn response). Further research would be necessary to directly link changes in arousal state of an individual to changes in head turn responses to repeated sound presentations, such as monitoring the heart rate. One example of changes in turning biases due to habituation has been reported in wombats (Descovich et al., 2013). Thereby, the habituation to the experimental set-up (without playback presentation) influenced the turning behaviour of the subjects, resulting in a change from left to right turning biases. The authors assumed that the initial left turning biases (i.e., without playback presentations) might be related to vigilance behaviour (Descovich et al., 2013). Concordantly, Zimmer & Demmel (2000) reported in humans that an initial right hemispheric activation in response to tone bursts diminishes due to habituation to repeated presentations of the sounds. These findings were discussed to be based on lateralised brain functions related to stress, emotions and motor preparation. This right hemispheric dominance in processing rather negative emotions, such as fear, anxiety and aggression has been found in several non-human vertebrate species (for review see Leliveld, Langbein & Puppe, 2013). Thus, in our study the initial high reactivity to the salient kitten calls (Aitkin, Tran & Syka, 1994; Konerding et al., 2016), might have led to a right hemispheric lateralisation that subsequently diminished during habituation to repeated sound presentations due to the decline in behavioural reactivity based on changes in the emotional state.

Ear position

Small deflections of the ear position significantly affected head turn direction with more same-side head turns than expected by chance. As the cat has highly and quickly movable pinnae (Populin & Yin, 1998), even an appropriate alignment of the body axis to the loudspeaker and the criteria that the ear position was parallel when the stimuli was played back did not guarantee that sound was perceived with similar loudness at both ears. The detailed frame-by-frame video analysis showed that within the reaction time of the experimenter the position of the pinnae slightly moved and only about 10% of head turn responses were confirmed to be preceded by a parallel orientation of the pinnae. When restricting the data set to these responses, we did not find a head turn asymmetry. However, due to the small sample size, a splitting of the remaining data set based on habituation was not feasible. Although same-side head turns occurred significantly more often than expected by chance, same-side head turns did not occur in all cases. Thus, a strong lateralised behaviour should not have been completely covered by the influence of the ear position. Still, this finding critically hampers the interpretation of the head turn paradigm as indicator of auditory laterality in cats, as mere sound level differences between the ears may have influenced turning responses (Heffner & Heffner, 1988).

General methodological considerations

The described influencing factors on the head turn response in cats, further strengthen the main criticism regarding the head turn paradigm (Fischer et al., 2009; Teufel, Ghazanfar & Fischer, 2010; Teufel, Hammerschmidt & Fischer, 2007): it has not been revealed that in a given subject an asymmetry in head turn behaviour serves as a reliable indicator for neuronal processes related to lateralised perception of an auditory stimulus (Teufel, Ghazanfar & Fischer, 2010). First, the neuronal origin of the head turn response has not been verified (Teufel, Ghazanfar & Fischer, 2010). Although, species-specific vocalisations have been described to asymmetrically activate auditory cortex regions both in humans (Zatorre, Belin & Penhune, 2002) and non-human primates (Poremba et al., 2004), the head turn response in cats may instead have a more subcortical origin (Beitel & Kaas, 1993; Teufel, Ghazanfar & Fischer, 2010) and might be related to emotional processing rather than auditory perception per se (Basile, Lemasson & Blois-Heulin, 2009; Reinholz-Trojan et al., 2012; Siniscalchi, Laddago & Quaranta, 2016; Xue et al., 2015). Mammalian infant cries, such as the kitten isolation call, have been described to activate specific subcortical regions, such as auditory thalamic areas and the amygdala, which are commonly activated during emotional processing (for review see Newman, 2007). Furthermore, the initial direction of unconditioned head orienting responses in cats has been found to be unaffected by bilateral ablation of the auditory cortices (Thompson & Masterton, 1978). Instead, it has been shown that the inferior colliculi of cats play an important role in reflexive orientation to sound in space (e.g., Syka & Straschill, 1970) which is based on inter-aural time and -level differences (Caird & Klinke, 1987). Second, it has been postulated that specialised brain regions, rather than whole hemispheres, exhibit the lateralisation that affects behavioural asymmetries (Wager et al., 2003). As brain regions might be lateralised independently from one another (Wager et al., 2003), the inference from an asymmetrical motor response to auditory lateralisation (at cortical level) is probably not as direct as assumed by the rationale of the head turn paradigm. Our results support these considerations, as both the arousal state (habituation) and subcortical processes (level differences between ears) influence the head turn behaviour. These findings complicate the interpretation of the paradigm, which generally assumes a turning response to the contralateral side of the dominant hemisphere.

Conclusion

We analysed for the first time potential auditory laterality in the domestic cat at individual-level. Based on repeated sound presentations, domestic cats did not show a turning asymmetry to kitten calls either at individual- or group-level, irrespective of call category or sex of the listener. When splitting the data set based on an observed habituation effect, we revealed an initial left ear advantage. This finding is discussed to be related to a right hemispheric dominance during induced high behavioural reactivity. As additional confounding factor, we revealed the ear/pinna position closer to the loudspeaker at sound onset. Due to the discussed limitations of the head turn paradigm, we highlight the importance of a comparative, neurological study to verify our results in the domestic cat. Our findings suggest that both the experimental sequence and the listening situation strongly influence the outcome of the head turn paradigm and should be taken into account when evaluating auditory laterality at behavioural level.

Supplemental Information

Figure S1 First head turn response to low and high arousal kitten call

Given are the individual values for males (blue) and females (red) in response to 25 playback presentations. The number of head turns is indicated to the right. Exact binomial test ∗p < 0.05.

Click here for additional data file.

Table S1 Raw data of head-turn responses

Given are the subjects, the sex, the affect, the sequence number of each playback presentation (order: 1–50), the ear position prior to listening to the call and the head turn direction (NA, no head turn).

Click here for additional data file.

We wish to thank K Möller for animal care and assistance during data collection, SVD Berg for technical support, Peter Baumhoff for preparing Figure 2 and B Haßfurther for video analysis for inter-observer reliability as well as F Sherwood-Brock for professionally proofreading the English.

Additional Information and Declarations

Competing Interests

Author Contributions

Animal Ethics

Data Availability

Elke Zimmermann is an Academic Editor for PeerJ.

Wiebke S. Konerding conceived and designed the experiments, performed the experiments, analyzed the data, wrote the paper, prepared figures and/or tables.

Elke Zimmermann conceived and designed the experiments, contributed reagents/materials/analysis tools, reviewed drafts of the paper.

Eva Bleich and Hans-Jürgen Hedrich contributed reagents/materials/analysis tools, reviewed drafts of the paper.

Marina Scheumann conceived and designed the experiments, analyzed the data, contributed reagents/materials/analysis tools, wrote the paper.

The following information was supplied relating to ethical approvals (i.e., approving body and any reference numbers):

The animal husbandry fulfilled all recommendations for domestic cats as required by the guidelines of the European Union (ETS 123, Directive 2010/63/EU) and was approved by the local veterinary authority (No. 42500/1H).

The following information was supplied regarding data availability:

The raw data has been provided as a Supplemental File.

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
