# Peer review of "The head turn paradigm to assess auditory laterality in cats: influence of ear position and repeated sound presentation"

_PeerJ, doi:10.7717/peerj.3925_

## Round 0.1 · original submission · Major Revisions

· Academic Editor

Major Revisions

As you can see the two Reviews varied somewhat in their evaluation. I mostly agree with the requests of Reviewer 1. I think you need to look at your data in some more detail, as the Reviewer suggested. For example, you found no significant response difference between the high versus low arousal calls. That is interesting but then, why not test other call that might reveal side differences?

Reviewer 1 ·

Basic reporting

109: Something is wrong with the grammar in this line/sentence.
Otherwise, the manuscript is written clearly and well. The references cited are adequate.
However, see my request to add more details on the method used, rather than slavishly referring the reader to previously published methods.

Experimental design

The authors rely too frequently on citation of previous papers that have used this method to score head turning. Instead, all of the details of the method should be given in this paper.

150-151: Say what is the evidence that the calls played back are of high versus low arousal since this is critical to interpreting the results. The sentence “The stimuli were distinct with regard to the arousal state of the sender….” does not make sense. Please clarify. Also, it is not clear what data have been tested statistically.

172: Here it is stated that the head had to be held “approximately straight”. This is too vague, please give the acceptable angles of the head.

174: Say precisely what position of the ears was considered to be “straight”.

Validity of the findings

231 and 232: What do the values n=7 and n=9 refer to here?

253 to 264: There was a significant group level bias to the left for the first 7 presentations but later it is said that the left turn bias did not differ significantly from chance. Does the latter statement refer only to the first head turn and, if so, how was a HTI value calculated?

Figure 3: Given that 7 of the 15 subjects shifted to an HTI of 1.0 in the second 7 presentations, it would be interesting to draw lines linking the score obtained in the first 7 presentations for each individual with the score for that individual in the following 7 presentations. These could then be tested using the paired Wilcoxon tests to see whether there has been a significant change at group level. Furthermore, since the data for the “following 7” look skewed, it would be better to show a median rather than a mean. Then a shift from left to right may be apparent and, at least, considered.

Additional comments

161: Correct ‘in details’ to ‘in detail’.

310-314: This interpretation of the results of Siniscalchi et al. (2008) is most unlikely to be the case because these authors looked at habituation and chose to analyse responses to the first 7 presentations since no significant habituation of responding occurred over the first 7 presentations (see their Figure 2).

I am positively inclined towards your examination of problems with the head turning paradigm but the manuscript would have beeen improved if you had included other calls that might have given different results, especially using a wider selection of calls that might elicit low versus high arousal responses.

·

Basic reporting

The paper is reasonably well written, although spelling and grammar require attention in quite a few places

Experimental design

No comment

Validity of the findings

No comment

Additional comments

This paper explores head turning in response to auditory stimulation at the individual level. The paper addresses an interesting and understudied area and draws attention to some important potential confounders in studies on this nature. I only have some small issues, highlighted below.


Abstract
This is well written, although inclusion of more methodological detail in places would certainly strengthen it.

Introduction
This is largely well written until the final few paragraphs, when the rationale for the study seems to become less clear. The authors start to talk about female cats (implying a sex effect?), but it is not quite clear whether they are referring to a sex effect or the study they mention only used female animals as its subjects. The last paragraph of the Introduction needs further expansion as it is confusing in nature and does not really set the reader up for the purpose of the investigation.


Spelling and Grammar
Line 30. Add ‘the’ before ‘individual’
Line 31. Add ‘the’ before ‘group’
Line 33. Change ‘influences’ to ‘influenced’
Line 43. Add ‘the’ before ‘behavioural’
Line 60. Change ‘over’ to ‘to’
Line 107. Change ‘will be….investigated’ to ‘would be…investigate’
Line 109. Remoe ‘these’ and change ‘shows’ to ‘show’
Line 161. Change ‘details’ to ‘detail’
Line 169. Change ‘maximal’ to ‘maximum’
Line 171 and 184. Change ‘to’ to ‘of’
Line 231. Remove ‘only’
Line 270. Change ‘on’ to ‘an’
Line 353. Change ‘indicators’ to indicator’
Line 358. Change ‘more a’ to ‘a more’

---

## Round 0.2 · Minor Revisions

· Academic Editor

Minor Revisions

The Reviewer is relatively satisfied with your revision but asks for a minor correction which will aid comprehension.

Reviewer 1 ·

Basic reporting

This has been improved.

Experimental design

No change but some extra details added. The authors' response included a photograph of the testing apparatus and that should be added to the paper itself to assist the reader in understanding the paper.

Validity of the findings

The findings are valid.

---

## Round 0.3 · accepted · Accept

· Academic Editor

Accept

It looks you have satisfactorily addressed all the requests of the Reviewers, and I am thus happy to accept your paper.